# Arsenic-Induced Carcinogenesis and Immune Dysregulation

**DOI:** 10.3390/ijerph16152746

**Published:** 2019-08-01

**Authors:** Hsin-Wei Huang, Chih-Hung Lee, Hsin-Su Yu

**Affiliations:** 1Department of Dermatology, Kaohsiung Chang Gung Memorial Hospital and Chang Gung University College of Medicine, Kaohsiung 833, Taiwan; 2Graduate Institute of Medicine, College of Medicine, Kaohsiung Medical University, Kaohsiung 807, Taiwan

**Keywords:** arsenic, carcinogenesis, Bowen’s disease, drinking water

## Abstract

Arsenic, a metal ubiquitously distributed in the environment, remains an important global health threat. Drinking arsenic-contaminated water is the major route of human exposure. Exposure to arsenic contributes to several malignancies, in the integumentary, respiratory, hepatobiliary, and urinary systems. Cutaneous lesions are important manifestations after long-term arsenic exposure. Arsenical skin cancers usually herald the development of other internal cancers, making the arsenic-induced skin carcinogenesis a good model to investigate the progression of chemical carcinogenesis. In fact, only a portion of arsenic-exposed humans eventually develop malignancies, likely attributed to the arsenic-impaired immunity in susceptible individuals. Currently, the exact pathophysiology of arsenic-induced carcinogenesis remains elusive, although increased reactive oxidative species, aberrant immune regulations, and chromosome abnormalities with uncontrolled cell growth might be involved. This review discusses how arsenic induces carcinogenesis, and how the dysregulated innate and adaptive immunities in systemic circulation and in the target organs contribute to arsenic carcinogenesis. These findings offer evidence for illustrating the mechanism of arsenic-related immune dysregulation in the progression of carcinogenesis, and this may help explain the nature of multiple and recurrent clinical lesions in arsenic-induced skin cancers.

## 1. Introduction

Arsenic is one of the most common metalloids on earth. Due to its ubiquitous nature, it poses a significant global health threat. Its name derives from arsenikon, the Greek name of yellow pigment. Based on the periodic table of chemistry, arsenic has been classified in the same group that includes nitrogen and phosphorus, both of which are essential chemical elements in cells [1]. The typical physical-chemical specific feature of arsenic to interact with biological tissue may result in its various and significant biological effects. Exposure to arsenic results in cancers of several human systems, such as integumentary, respiratory, hepatobiliary, and urinary systems [2]. In addition, arsenic exposure may contribute to the occurrence of multiple atherosclerotic vascular diseases, such as cerebrovascular events, myocardial infarction, and peripheral vascular diseases [3]. On the other hand, arsenic does have some benefits in treating specific diseases, such as lymphoma and leukemia. For example, arsenic is the drug of choice in patients of acute promyelocytic leukemia due to its biological effects in blast cell differentiation and immune cell activation [4].

Environmental exposure to arsenic can result from natural or anthropogenic routes. It enters the human body in several ways, such as oral ingestion, respiration, or skin absorption [5,6]. Oral ingestion with arsenic-contaminated water is the most common source [5]. To date, there remain more than 100 million people exposed to arsenic at levels higher than 50 µg/L through drinking water or via industrial sources [7]. For example, several decades ago in Taiwan, the residents of southwestern coastal areas often drank arsenic-contaminated groundwater and developed arsenic-induced cancers and vascular diseases several decades after arsenic exposure [8]. In addition, industrial exposure may also contribute to hazardous health problems. In 2014, the estimated worldwide production of arsenic was about 45,000 tons, most of them from China [9].

Chronic arsenic exposure induces several diseases [10], including cerebral vascular diseases [11], cardiovascular diseases [12], peripheral vascular diseases [8], and are associated with several infections, including pneumonia, hepatitis B infections, fungal skin infections [13,14,15].

Cutaneous manifestation often presents after exposure to arsenic about more than 20 years [16]. However, not all people exposed to arsenic develop skin cancers, suggesting that the dysregulated immunity caused by arsenic in susceptible patients may contribute to the progression of arsenic-induced cancers.

## 2. Skin Manifestations and Skin Cancer due to Chronic Arsenic Exposure

Arsenic leads to profound effects on many organs [17]. In addition to its direct toxicity, arsenic is a class I carcinogen as declared by the International Agency for Research on Cancer (IARC), and its carcinogenic effects may be mediated by abnormal DNA repair, aneuploidy, and other cellular mechanisms [18]. Skin is considered to be the most susceptible organ, and it is usually the first manifestation of arsenic toxicity [19]. In addition, only a certain proportion, ranging from 17–66%, of individuals exposed to arsenic contract cutaneous diseases, such as hyperpigmentation, hypopigmentation, Bowen’s disease, and arsenic keratosis. Furthermore, only around 1% of the individuals develop squamous cell carcinoma, basal cell carcinoma, or other cutaneous malignancies [20]. Individual immune dysregulation might explain why only a certain proportion of susceptible individuals develop adverse health effects.

Bowen’s disease, as squamous cell carcinoma in situ, is the most common skin cancers induced by arsenic. Clinically, typical Bowen’s disease is mainly associated with sun exposure and tends to be solitary, whereas arsenic-induced Bowen’s disease (As-BD) is distributed in sun-protected skin and tends to be multifocal [20]. However, there is no morphological discrepancy between classical and arsenic-induced Bowen’s disease in their pathological features, which include full-layered epidermal dysplasia, increased epidermal thickness, individual dyskeratosis, and moderate dermal infiltrates [20]. Variegated hyperpigmentation and punctate keratosis on palms and soles are the most significant clinical characteristics of arsenic-induced skin lesions [19,21]. These clinical features are often used as diagnostic clues to indicate chronic arsenic exposure. In addition, mucous membrane melanosis and conjunctiva congestion are minor dermatological presentations caused by exposure to arsenic.

Arsenic contributes to the formation of two important categories of non-melanoma skin cancers (NMSC), i.e., basal cell carcinoma (BCC) and squamous cell carcinoma (SCC). A high prevalence of NMSCs due to chronic arsenic exposure was noticed in the southwest of Taiwan, where the entire prevalence rate for cutaneous malignancies was 10.6/1000 [22]. Additionally, drinking arsenic-contaminated water has also contributed to arsenic-induced skin malignancies in Bangladesh, China, and India [16,23,24]. The association between inorganic arsenic in drinking water and miscellaneous malignancies has been well-confirmed after exposure to high concentration [25]. However, the epidemiologic evidence has suggested a threshold for the connection between inorganic arsenic exposure and malignancies [12]. Based on the epidemiological evidence mainly generated from the field study in Taiwan, in 2001, United States Environmental Protection Agency (USEPA) and World Health Organization (WHO) raised a more strict criterion for arsenic levels in drinking water, by decreasing the acceptable arsenic level in drinking water from 50 to 10 μg/L [26].

## 3. Pathophysiological Mechanisms of Arsenic-Induced Carcinogenesis

The precise pathophysiological mechanism by which arsenic induces carcinogenesis remains unclear, although increasing of oxidative stress, chromosome abnormalities, with uncontrollable growth, and aberrant immune developments, might be possible mechanisms of arsenic-induced carcinogenesis [9]. First, 8-Hydroxy-2-deoxyguanosine (8-OHdG), a form of reactive oxygen species (ROS) and a major form of oxidative DNA damage, was obtained from urine and skin tissue of arsenic exposed individuals [27]. Early genetic effects including DNA strand breaks, micronuclei in cord blood, and nitrative DNA damage were found in arsenic exposed patients [28]. In addition, oxidative damages and mitochondrial mutations might play a role in arsenic-induced skin malignancies [29,30]. Arsenic affects DNA repair machinery, which leads to oxidative DNA damage and mutations by the impairment of nucleotide excision repair, DNA ligase, DNA base excision repair, and DNA strand break rejoining [31,32]. Rather than the unrecoverable DNA damage, arsenic also affects a bunch of epigenetic regulations. For example, Chanda et al. showed that DNA hypermethylation of the critical promoter region of the p53 gene and p16 gene was present in the DNA from arsenic-exposed individuals [33]. Zhou et al. suggested a possible mechanism by which carcinogenesis is induced by arsenic via the dysregulated histone methylations that are involved in gene silencing and activating marks [34]. Liao et al. showed that un-methylation at −56 and −54 bp CpG in the cyclin D1 promoter as a predictor for invasive malignancies from carcinoma in situ in individuals with arsenic-induced cancers [35]. It was also reported that arsenic could regulate the PI3K/AKT/mTOR pathway to mediate cell growth [36,37].

In As-BD lesions, keratinocytes (KC) are the significant biological target. Abnormal cell proliferation and dysregulated energy homeostasis of keratinocytes are important in the mechanism and development of As-BD lesions. Lee et al. reported that arsenic can regulate the prolongation of glycan residues of membrane glycoproteins, which may be crucial in carcinogenesis induced by arsenic [1]. Besides, mtTFA up-regulation, increased mitochondrial biogenesis, and augmented mitochondrial functions induce cell proliferation in arsenic-induced skin cancers, indicating that mitochondria become a significant part in the induction of proliferation [30]. Lee et al. further showed that oxidative destruction and mutations in mtDNA might contribute to arsenic skin malignancy [29]. Moreover, expression of cytokeratin 14 (CK14) and the N-terminal truncated p63 isoform (ΔNp63, proliferation regulator) enhanced in As-BD, which contributes to abnormal growth in arsenic-induced skin malignancies [38]. Recent studies showed that epigenetic dysregulation may play an important role in arsenic-induced skin carcinogenesis [39]. Wu et al. revealed overexpression of has-miR-186 contributed to chromosomal instability in arsenic exposed tissue, including dysregulate chromatid segregation and induced aneuploidy [36].

## 4. Immune Response in Arsenic-Induced Skin Cancers and its Involvement in Carcinogenesis

As discussed earlier, not all of the humans exposed to arsenic develop cutaneous malignancies, suggesting that dysregulated immunity caused by arsenic in susceptible individuals contributes to the development of arsenic-induced cancers.

### 4.1. Overview of Skin Immunity

The human immune system includes two different functional groups: innate and adaptive immunity [40]. These two classifications have distinct characteristics of recognition receptors and differ in the speed at which they respond to a possible threat to the host. The innate immune systems include macrophages and dendritic cells (DCs), which lead to a rapid, but not lasting, response against the pathogens [41]. On the other hand, T and B lymphocytes were included in the adaptive immune system, which bears particular antigen receptors encoded by rearranged genes of T cell receptors and immunoglobulins [42]. Adaptive immunity develops more slowly compared to that of innate immunity. It can create and sustain memory of an immune reaction. Thus, it can also provide a rapid and robust reaction facing immunologic challenge. Based on the cell engagement and antibody production, the innate and adaptive immunity could be both categorized into cellular or humoral immunity, respectively [43].

### 4.2. Arsenic-Induced Dysregulated Immune Responses

As-BD, the most common carcinoma in situ due to chronic arsenic exposure, is featured with multiple and recurrent cutaneous manifestations. Besides, it may cause impairment of immunity in susceptible patients [35]. In mice, the production of immunoglobulins, proliferation of T lymphocytes, phagocytosis of macrophages, and their nitric oxide release ability were impaired in arsenic-exposed mice [20]. In zebrafish, Nayak et al. demonstrated that arsenic suppresses the overall innate immune system at an acceptable arsenic concentration in drinking water [44]. Monocytes and macrophages are also potential targets of arsenic. Lemarie et al. illustrated that an inorganic trivalent form of noncytotoxic level of arsenic trioxide (As_2_O_3_), significantly impairs proliferation of human blood monocyte-derived macrophages in vitro. It was also reported that arsenic induces rapid cell rounding and loss of actin reorganization, mostly via the Ras homolog gene family member A-associated kinase pathway [45]. Other studies have also reported that As_2_O_3_ significantly deteriorates the releasing of IL-12 and IL-23 from activated individual dendritic cells and decrease the activation of T-helper (Th) cells [46,47]. Moreover, an impaired macrophage function and delayed-type hypersensitivity response were also presented in arsenic-exposed individuals with skin manifestations [48].

In mice, transplacental studies of arsenic revealed that the offspring had a dose-dependent effect of the number of lung, liver, ovary, and adrenal tumors in adulthood [49]. Maternal exposure of arsenic augments oxidative stress in the placenta. In addition, pro-inflammatory cytokines IL-1, TNF-, and IFN- were increased through the enhancement of oxidative stress [50]. Lymphocyte function deteriorated and may contribute to the smaller size of thymic and decreasing function in newborns and infants [51]. In children, it is reported that arsenic exposure decreases the Th1 cytokine levels in the plasma and increases total concentrations of IgG and IgE in the plasma [52]. All of these immune deregulations could associate with the enhanced risk of immune dysregulation and infections in arsenic-exposed children. In adults, chronic arsenic exposure may lead to the deterioration of macrophage ability and impairment of peripheral blood polymorphonuclear leukocytes [53].

In human patients with arsenical cancers, impaired immunity does not only occur systemically, but also locally in As-BD skin lesions. Our group has shown that arsenic causes selective apoptosis of circulating CD4^+^ cells systemically in the peripheral blood. Once the surviving CD4^+^ cells are infiltrated into the local As-BD skin lesion, the soluble FasL from lesional keratinocytes triggers further CD4^+^ cell apoptosis through its binding to Fas in CD4^+^ cells. Moreover, we have also reported that vascular endothelial growth factor (VEGF) from cancer cells impairs activation and antigen presentation of Langerhans cell (LC) along with dysregulated T-cell activation in As-BD lesions [46]. Abnormal immune activation of LC and CD4^+^ cells provides a plausible mechanism for the impaired tumor surveillance in As-BD (Figure 1).

## 5. Conclusions

Arsenic is a ubiquitous element on the Earth. Due to its characteristic chemical and physical features, its biological and health effects are profound. Chronic arsenic exposure leads to many cancers in susceptible populations. Skin cancer is the most common and earliest arsenical cancer. The pathognomonic feature of arsenical skin cancer featuring abnormal epidermal keratinocyte proliferation, differentiation, dysplasia, and abnormal dermal inflammatory infiltrates may involve mitochondrial regulations associated with energy generation, ROS production, cell proliferation, DNA damage, and mutations, as well as immune regulations. The dysfunctional and impaired immune regulation might attribute to the differential susceptibility in arsenic-exposed humans. The impaired systemic and regional immune responses in arsenical carcinogenesis make it as a good model for early chemical carcinogenesis.

## Figures and Tables

**Figure 1 ijerph-16-02746-f001:**
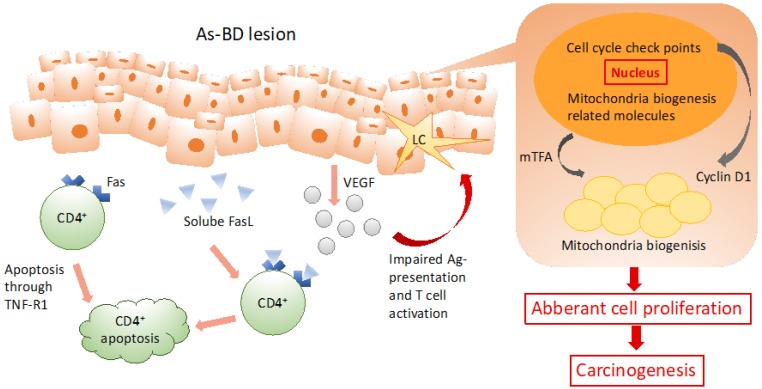
Aberrant immune activation of Langerhans cells (LCs) and CD4^+^ cells provides a reliable illustration for the impairment of anti-tumor surveillance in As-BD. In As-BD lesions, keratinocytes (KC) are the significant target. Dysregulated energy homeostasis and aberrant cell proliferation present as crucial role in arsenic-induced carcinogenesis.

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
