# Peer review of "Arsenic-Induced Carcinogenesis and Immune Dysregulation"

_ijerph, 2019, doi:10.3390/ijerph16152746_

Round 1

Reviewer 1 Report

This MS focus on the problem of skin cancer induced by exposure to arsenic. The English language should be improved, avoiding repetitions of words in the same sentence. References are required concerning As induced diseases. For example As induced skin cancers are missing. It is suggested to update the tumor effects induced by arsenic, in particular on the skin. However, there is a well description on arsenic-induced dysregulation on immune responses.

Other comments:

Page 1, Lines 10, 27 - Some imprecisions were noted (eg. Arsenic is a metaloid).

Lines 13-15 – Please rephrase the sentence: “Cutaneous lesions …carcinogenesis.”

Line 30 – Please note that arsenic is not essential to cells.

Line 32 – Please rephrase since some systems were mentioned as well organs.

Line 35 – arsenic instead of rsenic

Line 40 – Some references are required.

Page 1, Lines 30-35 and Line 9 Page 2. Please avoid repetition concerning biological effects of arsenic

Page 2, Line 20 – A reference is required.

Page 2, Line 21 – Which means solitary?

Please consider the following papers:

Barajas-Olmos FM, et al., Analysis of the dynamic aberrant landscape of DNA methylation and gene expression during arsenic-induced cell transformation. Gene. 2019 Jun 23:143941. 

 Wu J,et al., Overexpression of hsa-miR-186 induces chromosomal instability in arsenic-exposed human keratinocytes. Toxicol Appl Pharmacol. 2019 

Navasumrit P, et al., Exposure to arsenic in utero is associated with various types of DNA damage and micronuclei in newborns: a birth cohort study. Environ Health. 2019 Jun 7;18(1):51. 

Sawada N et al. Association between Arsenic Intake and Cancer-From the Viewpoint of Epidemiological Study]. Nihon Eiseigaku Zasshi. (2018)

Khairul I et al. Metabolism, toxicity and anticancer activities of arsenic compounds. Oncotarget. (2017)

Author Response

Dear Editors and Reviewers,

Thanks for your efforts for reviewing our manuscript and giving valuable comments.  We have revised the manuscript accordingly.  Please see our point-to-point responses below.

Point 1: This MS focus on the problem of skin cancer induced by exposure to arsenic. The English language should be improved, avoiding repetitions of words in the same sentence. References are required concerning As induced diseases. For example As induced skin cancers are missing. It is suggested to update the tumor effects induced by arsenic, in particular on the skin. However, there is a well description on arsenic-induced dysregulation on immune responses.

Response 1:

This previous version of manuscript has in fact been edited by the Language service provided by MDPI. Nevertheless, this revised version is edited by a native English speaker with several years of professional English editing service. 

We have added references for the arsenic induced disease.  See Page 2, Line 6-8.

“Chronic arsenic exposure induces several diseases [10], including cerebral vascular diseases [11], cardiovascular diseases [12], peripheral vascular diseases [8], and are associated with several infections, including pneumonia, hepatitis B infections, fungal skin infections [13-15].”

The references of arsenic induced skin cancers are incorporated too. See Page 2, Line 23-30.

“Bowen’s disease, as squamous cell carcinoma in situ, is the most common skin cancers induced by arsenic. Clinically, typical Bowen’s disease is mainly associated with sun exposure and tends to be solitary, whereas arsenic-induced Bowen’s disease (As-BD) is distributed in sun-protected skin and tends to be multifocal [20]. However, there is no morphological discrepancy between classical and arsenic-induced Bowen’s disease in their pathological features, which include full-layered epidermal dysplasia, increased epidermal thickness, individual dyskeratosis, and moderate dermal infiltrates [20]. Variegated hyperpigmentation and punctate keratosis on palms and soles are the most significant clinical characteristics of arsenic-induced skin lesions [19,21].”

Although the arsenic skin carcinogenesis is described in the section 2 and 3, we elaborated more pathologic mechanisms of arsenical carcinogenesis in skin deeper.  Please refer to Page 3, Line 14-15 and 26-29.

“It was also reported that arsenic could regulate the PI3K/AKT/mTOR pathway to mediate the cell growth [36,37].”

“Recent studies showed that the epigenetic dysregulation may play an important role in arsenic-induced skin carcinogenesis [39]. Wu et al. revealed overexpression of has-miR-186 contributed to chromosomal instability in arsenic exposed tissue, including dysregulate chromatid segregation and induced aneuploidy [36].”

Point 2: Page 1, Lines 10, 27 - Some imprecisions were noted (eg. Arsenic is a metaloid).

Response 2:

We have already revised the sentences as Page 1, Line 11 and 29.

“Arsenic, a metal ubiquitously distributed in the environment, remains an important global health threat.”

“Arsenic is one of the most common metals on earth.”

Point 3: Lines 13-15 – Please rephrase the sentence: “Cutaneous lesions …carcinogenesis.”

Response 3:

We have already revised the sentences as Page 1, Line 14-17.

“Cutaneous lesions are the important manifestations after long-term arsenic exposure. Arsenical skin cancers usually herald the development of other internal cancers, making the arsenic-induced skin carcinogenesis a good model to investigate the progression of chemical carcinogenesis.”

Point 4: Line 30 – Please note that arsenic is not essential to cells.

Response 4:

We have already revised the sentences as Page 1, Line 31-32.

“Based on the periodic table of chemistry, arsenic has been classified in the same group that includes nitrogen and phosphorus, both of which are essential chemical elements in cells [1].”

Point 5: Line 32 – Please rephrase since some systems were mentioned as well organs.

Response 5:

We have already revised the sentences as Page 1, Line 34-35.

“Exposure to arsenic results in cancers of several human systems, such as integumentary, respiratory, hepatobiliary, and urinary systems [2].”

Point 6: Line 35 – arsenic instead of rsenic.

Response 6:

We have already revised the sentences as Page 1, Line 37-38.

“On the other hand, arsenic does have some benefits in treating specific diseases, such as lymphoma and leukemia”.

Point 7: Line 40 – Some references are required.

Response 7:

Reference 5 and 6 had been quoted as Page 1, Line 41-43.

“Environmental exposure to arsenic can result from natural or anthropogenic routes. It enters human body through several ways, such as oral ingestion, respiration or skin absorption.[5,6] Oral ingestion with arsenic-contaminated water is the most common source [5].”

Point 8: Page 1, Lines 30-35 and Line 9 Page 2. Please avoid repetition concerning biological effects of arsenic

Response 8:

We have already revised the sentences as Page 2, Line 14.

“Arsenic leads to profound effects in many organs [17].”

Point 9: Page 2, Line 20 – A reference is required.

Response 9: reference 20 had been quoted as Page 2, Line 24-26.

“Clinically, typical Bowen’s disease is mainly associated with sun exposure and tends to be solitary, whereas arsenic-induced Bowen’s disease (As-BD) is distributed in sun-protected skin and tends to be multifocal [20].”

Point 10: Page 2, Line 21 – Which means solitary?

Response 10: Typical Bowen’s disease is mainly associated with sun exposure and tends to be solitary lesions, rather than multifocal distribution.

Point 11: Please consider the following papers:

1.Barajas-Olmos FM, et al., Analysis of the dynamic aberrant landscape of DNA methylation and gene expression during arsenic-induced cell transformation. Gene. 2019 Jun 23:143941.

 2.Wu J,et al., Overexpression of hsa-miR-186 induces chromosomal instability in arsenic-exposed human keratinocytes. Toxicol Appl Pharmacol. 2019

3.Navasumrit P, et al., Exposure to arsenic in utero is associated with various types of DNA damage and micronuclei in newborns: a birth cohort study. Environ Health. 2019 Jun 7;18(1):51.

4.Sawada N et al. Association between Arsenic Intake and Cancer-From the Viewpoint of Epidemiological Study]. Nihon Eiseigaku Zasshi. (2018)

5.Khairul I et al. Metabolism, toxicity and anticancer activities of arsenic compounds. Oncotarget. (2017)

Response 11:

1.Reference 39 had been quoted as Page 3, Line 26-27.

“Recent studies showed that the epigenetic dysregulation may play an important role in arsenic-induced skin carcinogenesis [39].”

2.Reference 36 had been quoted as Page 3, Line 14-15 and Line 27-29.

“It was also reported that arsenic could regulate the PI3K/AKT/mTOR pathway to mediate the cell growth [36,37].”

“Wu et al. revealed overexpression of has-miR-186 contributed to chromosomal instability in arsenic exposed tissue, including dysregulate chromatid segregation and induced aneuploidy [36].”

3. Reference 28 had been quoted as Page 3, Line 2-3.

“Early genetic effects including DNA strand breaks, micronuclei in cord blood, and nitrative DNA damage were found in arsenic exposed patients [28].”

4. Reference 6 had been quoted as Page 1, Line 41-42.

“Environmental exposure to arsenic can result from natural or anthropogenic routes. It enters human body through several ways, such as oral ingestion, respiration or skin absorption [5,6].”

5.Reference 47 had been quoted as Page 4, Line 9-11.

“Other studies have also reported that As2O3 significantly deteriorates the releasing of IL-12 and IL-23 from activated individual dendritic cells and decrease the activation of T-helper (Th) cells [46,47].”

Reviewer 2 Report

This is a review discussing the implication of arsenic-mediated dysregulation of the immune system in individuals ultimately developing arsenic-induced cancers.

The review is well structured, however it contains some grammar and syntax errors that need to be corrected (for examples, please refer to the uploaded file).

I feel that the authors could add some more information supported by the relevant bibliography in paragraph 4.1 describing skin immunity in general, before discussing the effects of arsenic on immune responses.

Author Response

Dear Editors and Reviewers,

Thanks for your efforts for reviewing our manuscript and giving valuable comments.  We have revised the manuscript accordingly.  Please see our point-to-point responses below.

Point 1: This is a review discussing the implication of arsenic-mediated dysregulation of the immune system in individuals ultimately developing arsenic-induced cancers.

The review is well structured, however it contains some grammar and syntax errors that need to be corrected (for examples, please refer to the uploaded file).

Response 1:

I had already revised the errors following the uploaded file and highlighted it. Thank you very much for your great opinions.

Point 2: I feel that the authors could add some more information supported by the relevant bibliography in paragraph 4.1 describing skin immunity in general, before discussing the effects of arsenic on immune responses.

Response 2:

Reference 40-43 had been quoted as Page 3, Line 36-45.

“The human immune system includes two different functional groups: innate and adaptive immunity [40]. These two classifications have distinct characteristics of recognition receptors and differ in the speed at which they respond to a possible threat to the host. The innate immune systems include macrophages and dendritic cells (DCs), which lead to a rapid but not lasting response against the pathogens [41]. On the other hand, T and B lymphocytes were included in the adaptive immune system, which bear particular antigen receptors encoded by rearranged genes of T cell receptors and immunoglobulins [42]. Adaptive immunity develops more slowly compared to that of innate immunity. It can create and sustain memory of an immune reaction; thus, it can also provide a rapid and robust reaction facing immunologic challenge. Either the innate or adaptive immune can be categories as cellular or humoral immunity, respectively [43].”

Round 2

Reviewer 1 Report

Authors have done several changes for the improvement of the MS.

Minor points are required:

Page 1, Line 29 – Arsenic is a metaloid

Page 3, Line 30 - 4. Immune response in arsenic-induced skin cancers and its involvement in carcinogenesis

Author Response

Dear Reviewers,

Thanks for your efforts for reviewing our manuscript and giving valuable comments.  We have revised the manuscript accordingly.  Please see our point-to-point responses below.

Reviewer 1

Point 1: Page 1, Line 29 – Arsenic is a metaloid

Response 1:

We have revised the sentence as that in Page 1, Line 29.

Arsenic is one of the most common metalloids on earth.”

Point 2: Page 3, Line 30 - 4. Immune response in arsenic-induced skin cancers and its involvement in carcinogenesis

Response 2:

We have revised the sentence as than in Page 3, Line 31.

“4. Immune response in arsenic-induced skin cancers and its involvement in carcinogenesis”
